# Pilot Investigation of Anti-*Salmonella* Antibodies in Oral Fluids from *Salmonella* Typhimurium Vaccinated and Unvaccinated Swine Herds

**DOI:** 10.3390/ani11082408

**Published:** 2021-08-14

**Authors:** Alessia De Lucia, Shaun A. Cawthraw, Richard Piers Smith, Rob Davies, Carlo Bianco, Fabio Ostanello, Francesca Martelli

**Affiliations:** 1Department of Veterinary Medical Sciences, School of Agriculture and Veterinary Medicine, via Tolara di Sopra 50, 40064 Ozzano Emilia, Italy; alessia.delucia3@studio.unibo.it; 2Animal and Plant Health Agency, Woodham Lane, New Haw, Addlestone KT15 3NB, UK; shaun.cawthraw@apha.gov.uk (S.A.C.); richard.p.smith@apha.gov.uk (R.P.S.); rob.davies@apha.gov.uk (R.D.); francesca.martelli@apha.gov.uk (F.M.); 3Animal and Plant Health Agency Lasswade, Pentlands Science Park, Bush Loan, Penicuik, Midlothian EH26 0PZ, UK; carlo.bianco@apha.gov.uk

**Keywords:** antibodies, ELISA, oral fluid, pigs, *Salmonella*

## Abstract

**Simple Summary:**

The consumption of pork meat is responsible for a significant number of outbreaks of salmonellosis in people. Surveillance in pig herds is constrained by the cost-effectiveness and efficiency of sampling methods. The last decade has seen significant advances in the routine use of pool samples, including oral fluids (OFs). This study aimed to investigate the OF collected passively via chewed sampling ropes as a potential sample type for assessing anti-*Salmonella* antibodies in two *Salmonella*-vaccinated (V) and two non-vaccinated (NV) farrow-to-finish pig farms, comparing the results with the *Salmonella* shedding of tested animals. Sows in the V farms were vaccinated prior to farrowing. Pooled faecal and OF samples were collected from sows and their offspring. *Salmonella* was isolated with direct bacteriological methods. A commercial ELISA assay was adapted to detect IgG and IgA antibodies in OF. Overall, a higher *Salmonella* prevalence was observed in the NV farm and in the offspring (76.3%) compared to sows (36.4%). The protocol used to test anti-*Salmonella* IgA in pig OF samples was found to lack sensitivity and specificity. At herd level, IgG is the most reliable isotype for monitoring *Salmonella* specific antibody via OF.

**Abstract:**

Oral fluid (OF) can be a simple, cheap and non-invasive alternative to serum or meat juice for the diagnosis and surveillance of important pathogens in pigs. This study was conducted on four *Salmonella* Typhimurium-positive farrow-to-finish pig farms: two *Salmonella*-vaccinated (V) and two non-vaccinated (NV). Gilts and sows in the V farms were vaccinated with a live, attenuated vaccine prior to farrowing. Pooled faecal and OF samples were collected from the sows and their offspring. *Salmonella* was isolated according to ISO6579–1:2017. In parallel, IgG and IgA levels were assessed in OF samples using a commercial ELISA assay. *Salmonella* was detected in 90.9% of the pooled faecal samples from the NV farms and in 35.1% of the pooled faecal samples from the V farms. Overall, a higher prevalence was observed in the pooled faecal samples from the offspring (76.3%) compared to the sows (36.4%). IgG antibodies measured in V farms are likely to be related to vaccination, as well as exposure to *Salmonella* field strains. The detection of IgA antibodies in OF was unreliable with the method used. The results of this study show that IgG is the most reliable isotype for monitoring *Salmonella*-specific antibody immunity in vaccinated/infected animals via OF.

## 1. Introduction

Non-typhoidal salmonellosis is regarded as one of the most important food-borne zoonotic diseases, causing ill health and high disease-related costs in people [1,2]. The consumption of pork meat is a major source of human outbreaks [3]. Pigs are susceptible to most *Salmonella* serotypes and, although Typhimurium and its monophasic variants (mST) are the most common, a large variety of other serotypes are also reported in surveillance studies at farm level [3].

To control the infection in pigs, a combined on-farm approach has been proposed: external and internal biosecurity, control of *Salmonella*-contaminated feed and vaccination. A live, attenuated vaccine against *S.* Typhimurium in pigs has been developed in Germany (Salmoporc—CEVA Animal Health, Libourne, France) and is currently available in some European countries.

Currently there is no legislation on the control of the *Salmonella* infection in live pigs. Diagnosis and surveillance for *Salmonella* in pigs can be carried out on farm or at slaughter using conventional culture methods or serological techniques [4].

Some European countries such Denmark and Germany established their national control programmes to determine the prevalence of *Salmonella*-positive pig herds on the basis of serological surveillance. Serological monitoring is performed on meat juice collected at the abattoir and tested for *Salmonella* antibodies using an ELISA. According to the serological status, farms are assigned to three epidemiological levels [5]. Highly infected herds are assigned to level two or three, and farmers are supported by the national governments to reduce the infection load of their herd. Additionally, these farms are subjected to penalty fees to cover the expenses of the special hygienic precautions that have to be taken at the slaughterhouse when pigs from herd level three are slaughtered [6,7]. Farmers are, therefore, motivated to apply better control measures to reduce *Salmonella* prevalence and avoid the financial consequences [6].

*Salmonella* surveillance in pig herds is constrained mainly by the cost-effectiveness and efficiency of sampling methods [8]. Disease monitoring often involves blood sampling or environmental samples (floor swabs for *Salmonella*), which are costly to the farmer due to veterinary fees and labour [4,8]. The monitoring of herd health on a regular basis offers accurate diagnostic information and provides options for intervention strategies that can be implemented during the animal’s lifetime.

Serological assays using oral fluid (OF) have recently been developed for veterinary diagnostics as OF examination may prove a useful and convenient diagnostic measure of group disease status in pigs [9]. Oral fluid is composed of saliva and a transudate that originates from oral capillaries, particularly gingival crevicular fluid that leaks from the crevices between the teeth and gums [10]. This transudate is a product of the circulatory system and, consequently, contains many of the components found in serum [9]. As such, OF has been described as a diagnostic “mirror of the body”, as antibodies from IgA, IgG and IgM classes are all present [11,12]. The major antibody class in saliva is secretory IgA (sIgA) produced by local plasma cells in the salivary gland. In contrast, the major class in crevicular fluid is IgG [13,14]. Antibodies of this class are derived from serum, although some IgG antibodies are also locally produced [15]. The presence of local and systemic antibodies in OF suggests they may be suitable for the immunodiagnosis of infectious diseases in live animals. The use of OF has several advantages compared to serum. Sample collection is relatively stress-free for the animals, and cheap and easy to perform, even by unskilled personnel [16,17]. Oral fluid offers the possibility of testing pooled samples that facilitates cost-efficient monitoring of the health status of a large population [11,18]. Many serum assays can be optimised to detect antibodies in OF [19] and a number of recent studies have investigated their potential for disease diagnosis in pigs [20,21,22].

Oral fluid can be obtained using ropes made from a range of natural fibres, such as cotton and hemp, and synthetic fibres, such as polyester and polyamide (water absorbing) or polypropylene and polyethylene (water repellent) [15]. Importantly, the rope material seems to have an impact on the antibody titre obtained and the isotypes of the antibodies collected. Cotton is highly absorbent and reportedly yields higher titres of IgG antibodies compared to other rope types [11,15]. Pig saliva has been used to detect antibodies against several specific porcine pathogens and is now routinely used for the virus isolation of important endemic swine pathogens such as PRRSV and PCV2 [21,23].

The objective of this study was to investigate specific anti-*Salmonella* IgG and IgA antibodies levels in OFs collected from pigs that were vaccinated or not vaccinated against *Salmonella* Typhimurium, in comparison with the shedding of *Salmonella* in pooled faecal samples of tested animals.

## 2. Materials and Methods

### 2.1. Farms

The samples were collected from four *Salmonella*-positive farrow-to-finish indoor farms in the UK, sampled within a previous research project aimed at evaluating the efficacy of a live *Salmonella* Typhimurium vaccine [24]. The following inclusion criteria were used: (i) indoor breeder-finisher enterprise, (ii) herd size of 100–700 sows, (iii) presence or recent occurrence of *Salmonella* Typhimurium or mST and (iv) sows free of significant clinical disease that may have affected the efficacy of the vaccine [24]. Two of the randomly selected farms used vaccination against *Salmonella* (V), while the other two did not (NV). In the V farms, the gilts and sows were vaccinated subcutaneously, at 6 weeks and 3 weeks prior to farrowing with Salmoporc STM (CEVA Animal Health, France). The sows received a booster vaccination three weeks before each farrowing. In all farms, the following three pig categories were sampled: farrowing sows and the offspring weaners (from 4 to 10 weeks) and grower pigs (from 11 to 15 weeks) (Figure 1).

One farm visit was carried out to each of the four farms within a three-month period. Sampling visits took place at a point where about half of the progeny on the vaccine farms were estimated to have originated from vaccinated sows. One year after the end of the study, all progenies came from vaccinated sows.

### 2.2. Oral Fluid and Pooled Faeces Collection

In order to see whether a commercial ELISA test validated for serum and meat juice (MJ) was able to accurately detect anti-*Salmonella* antibodies in OF, samples were obtained from different pig categories (farrowing sows, weaners, grower pigs). The sampling scheme was representative of the different production and age categories within farms. Three OFs were collected from first- or second-parity sows, four were collected from third- fourth-parity sows and three from the older sows. Regarding the offspring, samples were collected from weaners (about 4–10 weeks old) and growers (about 11–15 weeks old). In each farm and for each pig category, 10 samples of OF were collected. To serve as negative controls in addition to these farm samples, four OF samples were collected from *Salmonella*-free sows housed in biosecure pens at the Animal and Plant Health Agency (APHA), UK.

For OF sample collection, 50-centimeter-long cotton ropes were placed at pig shoulder height and left in pens of 25–30 pigs for 30 to 60 min, in order to allow approximately 75% of animals in the pen to chew the rope [23]. When group sizes were larger, one rope for each multiple of 30 pigs was hung in different areas within the same pen. Where there were multiple ropes in a single pen, each rope was treated as single sample rather than pooled, as pooling may influence diagnostic results. Ropes were then placed in individual plastic bags with minimal handling to avoid cross-contamination, transported chilled to the laboratory in less than four hours and refrigerated (+4 °C) overnight. The following day, OFs were extracted by squeezing the ropes and collected into tubes. All samples were centrifuged (4650× *g* per 10 min) and the supernatants stored in aliquots at −80 °C.

From each pig category, approximately 25 g pooled faecal samples were taken from the floor with a fabric hand swab and placed directly into 225 mL of buffered peptone water (BPW; Merck, Darmstadt, Germany, 1.07228.0500) [25].

### 2.3. Bacteriological Analyses of Salmonella Prevalence in Pooled Faecal Samples

*Salmonella* was isolated according to a modification of ISO6579-1:2017. Briefly, all inoculated BPW samples were incubated at 37 ± 1 °C for 16–20 h and, subsequently, 0.1 mL of each was inoculated onto modified semi-solid Rappaport-Vassiliadis (MSRV; Mast Group, Bootle, UK, DM440D, with addition of 1 mg/mL of novobiocin, Sigma-Aldrich, Darmstadt, Germany, N1628) enrichment agar and incubated at 41.5 ± 1 °C for 24 ± 3 h. Growth on MSRV was sub-cultured onto Rambach agar (Merck, Darmstadt, Germany, 1.07500.0002), which was incubated at 37 ± 1 °C for 24 ± 3 h. Serotypes were determined for all isolates according to the White–Kauffmann–Le Minor scheme [26].

### 2.4. Detection of Anti-Salmonella IgG and IgA in OF samples

Anti-*Salmonella* IgG antibodies present in the OF were measured using the IDEXX Swine *Salmonella* Ab Test (IDEXX Laboratories, Westbrook, ME, USA), which has been validated for serum and MJ but not for OF. A previous study has shown that under field conditions and at different cut-points of sera optical density (OD), this ELISA kit has a sensitivity of 29–53% and specificity of 72–93% when compared with faecal culture results [27]. Always according to the cut-points of OD values, results of MJ examination with this ELISA kit have a relative sensitivity of 53–92% and relative specificity of 64–84% when results from serum ELISA were used as gold standard [28]. The manufacturer’s protocol was followed except that the OF samples were tested (diluted 1:1 in dilution buffer) instead of serum or MJ. With these changes, previous results indicate that saliva samples had a sensitivity and specificity of 86 and 80%, respectively, when compared with ELISA results obtained from individual serum samples [29].

The IDEXX ELISA plates were also used to detect *Salmonella*-specific IgA in OF samples. The protocol outlined above was followed except that the kit conjugate was replaced with an anti-porcine IgA HRP conjugate (Abcam, Cambridge, UK), used at 1:10,000 dilution. The positive and negative kit controls and OF collected from four *Salmonella*-free sows housed in biosecure pens at APHA were included on each plate. In order to account for variation within the assay, all samples were tested in duplicate, and the coefficient of variation (CV) was calculated.

### 2.5. Statistical Analyses

The Kolmogorov–Smirnov test (K–S) for goodness of fit was used to verify normality of the anti-*Salmonella* IgG and IgA OD values. According to the K–S test results, the Mann–Whitney (M–W) U-test was used to compare anti-*Salmonella* IgG and IgA OD values between V and NV farms and between production and age categories. Fisher’s chi-square test was used to compare prevalence of *Salmonella* isolation in pooled faecal samples between farm categories (V, NV) and between production categories (sows and offspring). To measure how strongly *Salmonella* positivity in pooled faecal samples was associated with the absence of vaccination, odds ratio (OR) and 95% confidence intervals (95%CI) were calculated. To assess a possible correlation between *Salmonella* prevalence in pooled faecal samples and anti-*Salmonella* IgG and IgA OD values, the nonparametric Spearman’s *rho* correlation coefficient was calculated.

Statistical significance was set at *p* ≤ 0.05. All statistical analyses were performed using the software SPSS 25.0.0 (IBM SPSS Statistics, Armonk, NY, USA).

## 3. Results

### 3.1. Detection of Anti-Salmonella IgG and IgA in OF Samples

A total of 120 OF samples were collected, but 21 were discarded as they were faecally contaminated and a further 18 were discarded as the ropes had not been chewed. The 81 OF samples that were tested originated from V farms (37) and NV farms (44) (Table 1).

The volume of OF obtained from the ropes ranged from 2–10 mL (except for one sample that was only tested for anti-*Salmonella* IgA). The intra-assay CV (CV _intra-assay_) was calculated as the average of the individual CVs. The CV was between 0 and 20.8% (CV _intra-assay_: 4.15%) and between 0 and 30.9% (CV _intra-assay_: 7.3%) for IgG and IgA, respectively. Usually, a CV _intra-assay_ of 10% or less is considered satisfactory [30].

The median, minimum and maximum ELISA OD values detected in OF are shown in Table 2.

Considering all the pig categories, the assays for IgG detected a significant higher anti-*Salmonella* IgG OD value in the OF samples of the V farms (M–W U = 470.5; *p* = 0.002) (Figure 2). In contrast, there was no significant difference between the anti-*Salmonella* IgA OD values between the V and NV farms (M–W U = 799.5; *p* = 0.887) (Figure 2).

In both the V and NV farms, there was no significant difference between the anti-*Salmonella* IgG in the OF of sows and their offspring (M–W U = 131.0; *p* = 0.558 and M–W U = 108.0; *p* = 0.156, respectively) (Figure 3).

Only the sows in the V farms had significantly higher IgA levels than their offspring (M–W U = 83.0; *p* = 0.02) (Table 2).

### 3.2. Bacteriological Results of Salmonella Prevalence from Pooled Faeces

Eighty-one pooled faecal samples were collected using a hand-held gauze; 37 from the V and 44 from the NV farms. Details from the 81 bacteriological faecal pool samples examined are presented in Table 3.

In the two V farms, from the 37 faecal samples taken, *Salmonella* was recovered from 13 samples (35.1%). Of these, two were S. Typhimurium and 11 were mST. From the 44 faecal samples collected in the NV farms, *Salmonella* was recovered from 40 samples (90.9%). mST was also isolated from both of these farms (22 of 40 positive samples), although in one of these farms, *S.* Kedougou was more prevalent (18 of 40) (Table 3).

In the NV farms, there was a significantly higher (*p* < 0.001) *Salmonella* prevalence of pooled faecal samples than in the V farms (90.9% vs. 35.1%) (Table 3). Single-factor-analyses revealed statistically significant associations between *Salmonella* positivity in pooled faecal samples and the absence of vaccine use (OR: 18.5; 95%CI 5.4–63.1).

Similarly, in the NV farms, the prevalence of both sows and offspring was significantly higher than in the V farms (*p <* 0.05 and *p* < 0.001, respectively), (Table 3).

A significant (*p* = 0.001) higher *Salmonella* prevalence was detected in samples collected from the offspring (45/59; 76.3%) when compared to the sows (8/22; 36.4%).

In the NV farms, the *Salmonella* prevalence of both sows and offspring and of all the pig categories was not significantly different (*p* > 0.05) between the two farms (3 and 4), (Table 3).

In the V farms, a statistically significant difference in the *Salmonella* prevalence in samples collected from the offspring (*p* < 0.001) and from all the pig categories (*p* < 0.01) was observed between the two farms (1 and 2). On the contrary, the prevalence in sows was not significantly different (*p* > 0.05).

### 3.3. Correlation between Salmonella Prevalence in Pooled Faecal Samples and Anti-Salmonella IgG and IgA OD Values

In the V farms, when the median values of anti-*Salmonella* IgG and IgA were compared with *Salmonella* prevalence using the Spearman’s *rho* coefficient, no correlation was observed (*rho*: 0.121; *p* = 0.484 and *rho*: 0.268; *p* = 0.108, respectively).

On the contrary, in the NV farms, a positive correlation was observed between the median of the anti-*Salmonella* IgG and IgA level and *Salmonella* prevalence (*rho*: 0.358; *p* = 0.017 and *rho*: 0.0632; *p* < 0.001, respectively).

## 4. Discussion

This study aimed to investigate the levels of anti-*Salmonella* IgG and IgA antibodies in OF in pigs from herds vaccinated or not vaccinated with a live, attenuated *Salmonella* vaccine. In addition, the bacteriological status of pooled faecal samples was also determined.

Antibody levels were determined using a commercially available ELISA (IDEXX Swine *Salmonella* Ab Test) that has been validated for the detection of *Salmonella* antibodies in porcine serum and MJ. The ELISA kit used in this study has also been used to test porcine OF for anti-*Salmonella* antibodies in a previous study [21,31]. Recently, the correlation of anti-*Salmonella* antibodies between serum, OF and saliva samples collected from pigs was evaluated using this commercial kit [29]. The assay screens for the presence of antibodies to the most commonly occurring *Salmonella* serogroups (B, C1, D) in pigs. *S.* Typhimurium and its variants, which belong to serogroup B, were recovered from the samples from each study farm. However, a group G serovar, *S.* Kedougou, was also isolated from one of the NV farms, more frequently than *S.* Typhimurium. Antibodies to this serovar may not be detected by the assay used in this study [32]. Furthermore, *S.* Kedougou is not normally considered an invasive serovar and, therefore, may induce a more moderate systemic antibody response. It has been suggested that serological testing may have a limited role in monitoring infection by non-invasive *Salmonella* serotypes [33]. However, anti-*Salmonella* antibodies were detected in the OF collected in this farm, suggesting that those animals had a mixed infection with *S.* Typhimurium and *S*. Kedougou. *Salmonella* co-infections are not rare in pig herds [34], but for this study, serotyping was performed on only one colony from each positive faecal sample; therefore, it is likely that only the predominant serovar was detected.

IgG is the most abundant isotype in blood and extracellular fluid, while IgA is predominant at mucosal surfaces and in secretions, including saliva. IgA antibodies are a first line of defence against pathogens, preventing the attachment of bacteria or toxins to epithelial cells [35]. Our results relating to anti-*Salmonella* IgG antibodies in OFs reflect observations from similar studies using serum as diagnostic material [36]. Specific salivary IgG antibodies have been previously reported as potential indicators of enteric infections [37,38], showing that OF anti-*Salmonella* IgG antibodies assessment may represent a simple, cheap and non-invasive alternative to serum or MJ. In this study, a positive correlation was observed between the anti-*Salmonella* IgA level and *Salmonella* prevalence in the NV farms. However, anti-*Salmonella* IgA detection was found to be unreliable and non-specific. Indeed, the IDEXX kit detected high anti-*Salmonella* IgA OD values in the OF of the four *Salmonella*-negative sows used as controls, suggesting a false positive and non-specific binding (Table 2). The use of cotton ropes seems to be appropriate and recommended to yield higher amounts of IgG. It has been reported that cotton fibre can result in lower IgA concentrations when compared with synthetic fibres [11]. However, results from a recent study showed strong anti-*Salmonella* IgA responses in OF samples collected with cotton rope [31]. There are key differences between this study and ours. The study by Atkinson et al. [31] was a controlled challenge study conducted under experimental conditions that differ considerably to the field conditions of our study. Controlled experimental conditions are presumably less stressful than those found in the intensive pig production units studied in our trial. The animals in a study conducted by Atkinson et al. [31] underwent a single experimental infection and were tested within a short time frame from the infection, whilst the pigs in our investigation were naturally and chronically infected by *Salmonella*. Furthermore, the specific, strong IgA responses were only detected using an in-house ELISA, using a whole cell antigen preparation made from the challenge strain.

The results from direct diagnostic methods (bacteriology) and indirect diagnostic methods (serology) may not necessarily correlate. The culture of *Salmonella* indicates true infection and transmission, whereas positive serology may also indicate latent infection within the herd or previous infection [39]. In our study, a lower *Salmonella* prevalence was detected in the V farms, in particular for the vaccinated sows (Table 3). However, an important variability was observed within the two farm categories (Figure 2 and Figure 3), especially within the V farms. In V farm 1, a lower *Salmonella* prevalence was found in sows compared with their offspring, as would be expected, while in V farm 2, the opposite was observed, with a higher prevalence found in the vaccinated sows (Table 3). There are several plausible explanations for the variability in the vaccine effect in terms of prevalence in the pig categories. Each pig herd is unique regarding biosecurity measures, management, location, facilities, host susceptibility and other influential factors, which can lead to the large variability within the two farm categories, V and NV [24]. Despite that, the odds ratio of *Salmonella*-positive samples was significantly higher in the NV compared to the V farms.

In the NV farms, a positive correlation was observed between the median of the anti-*Salmonella* IgG level and the *Salmonella* prevalence. The high *Salmonella* burden with high antibody levels in the offspring (in comparison with the sows) suggested that the antibody response was related to infection with field *Salmonella* strains. Differently, the significantly (*p* < 0.001) lower *Salmonella* prevalence in the V farms, as would be expected, is likely related to the higher antibody levels detected in the OFs of vaccinated sows, which decreases in their offspring due to the lacking vaccination booster (Figure 3). As already showed by a recent study, the high level of IgG antibodies in the OFs of vaccinated sows may arise from important systemic and mucosal humoral immune responses to anti-*Salmonella* vaccination [40]. Differences in IgG antibody OD values were also observed in the pig categories of the V and NV farms (Table 2 and Figure 3). In one of the V farms (farm 1), vaccine-induced responses are presumably, at least partly, responsible for the higher levels of IgG antibodies in the OF samples seen in the vaccinated sows compared to those in their non-vaccinated offspring (Table 2). In contrast, on the NV farms, higher IgG OD values were observed in the offspring (in comparison with the sows). This could indicate higher levels of current/recent infection in the younger animals, as supported by the bacteriological results (Table 3). It is important to consider that the OF were pooled, and a possible dilution effect should be taken into account. Previous studies expressed concern about the impact of dilution and incomplete group sampling when pooling saliva from pigs voluntarily chewing sampling ropes. When pooled samples are collected from a group of animals, the contribution of individual animals to the pool is unknown and, therefore, is suitable only to evaluate the current group status. Positive pools indicate that at least one individual sample within the pool is positive; therefore, it is necessary to retest each sample to decode the positive from the negative animals. On the contrary, it is possible that positive animals may not be included in the pool, leading to seropositive subjects being misdiagnosed by reducing the value below the cut-off point [23]. Modifications to the test protocol (e.g., increasing the test volumes, sample dilutions, incubation times, temperature, kit reagents and cut-off point) may be necessary to optimise the performance of the assay [19]. These modifications need to be evaluated and validated against the current Gold Standard (blood serum samples) for assessing the sensitivity and specificity of the assay. At present, only the MJ has been validated and accepted as a surveillance option for the detection of anti-*Salmonella* antibodies [41]. However, considering the cost-effective benefit of using the OF for diagnostic purposes, at farm level, a lower sensitivity may be offset, either by collecting a higher number of samples at each sampling point or by using routine surveillance testing (every 2–4 weeks) [23].

In porcine OF, antibody concentrations are much lower compared to serum. It was noted that IgG concentrations in OF are 800 times lower than in serum [11]. Despite the limitations of our study, such as the lack of a baseline for antibodies in the study herds and the lack of information regarding the course of infection and exposure, the results show that OF is a promising sample type for monitoring IgG levels in V farms.

The animals’ behaviour in relation to age is an important factor that should be taken into account as it can affect the success of the OF sampling and the results of the study. A number of ropes were discarded because the sows did not chew them; therefore, the number of OF samples collected from sows was lower than that of the samples collected from offspring. In growing pigs, their natural exploratory behaviour facilitates the collection of OF, whereas older animals, such as gestating sows or boars, are generally less curious and less motivated to explore materials [42,43]. For this reason, the collection of OF samples from sows is usually conducted on individually housed animals instead of on group-housed animals [42]. However, the training of sows by repeated exposure to the collection process seems to improve the animals’ interest to chew the device [23].

The anti-*Salmonella* IgG detection in OF appears to be a promising technique, as Decorte et al. [15] already showed, and IgG in OF is the most reliable immunological isotype for monitoring specific antibody immunity in vaccinated/infected pigs.

However, the results obtained must be evaluated with caution due to some limitations of the study: in fact, the number of herds and samples examined was relatively low and no information on the serological status of the animals was available. Moreover, pigs of different ages with different health statuses and different behaviours were evaluated.

## 5. Conclusions

The current study, focusing on OF anti-*Salmonella* IgG, provides a preliminary indication of the potential value of this sample type on pig farms. On-farm surveillance of zoonotic diseases at the pre-harvest level is useful to help anticipate a public health problem long before it escalates. In this context, OF sampling represents a promising approach to meet this objective. The detection of specific anti-*Salmonella* IgG in OF could allow for the detection of *Salmonella*-positive batches of pigs and the adoption of specific prophylaxis measures at the slaughterhouse (e.g., end-of-day slaughter of high prevalence batches). However, further studies are needed to evaluate the correlation between *Salmonella* isolation and the presence of specific IgG in animals, in order to establish an effective monitoring protocol on farms. Further studies are necessary to confirm and expand our findings. It is, therefore, recommended that further larger scale studies are carried out in order to give greater confidence in the reliability of the method. A larger study may have enabled the results to be modeled directly against the *Salmonella* prevalence and count data, which might have produced more biologically relevant results.

## Figures and Tables

**Figure 1 animals-11-02408-f001:**
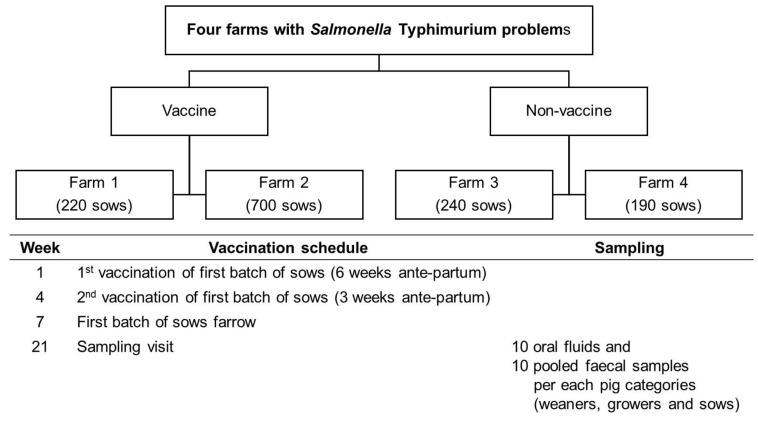
Experimental design: schedule of the vaccination programme and sampling scheme.

**Figure 2 animals-11-02408-f002:**
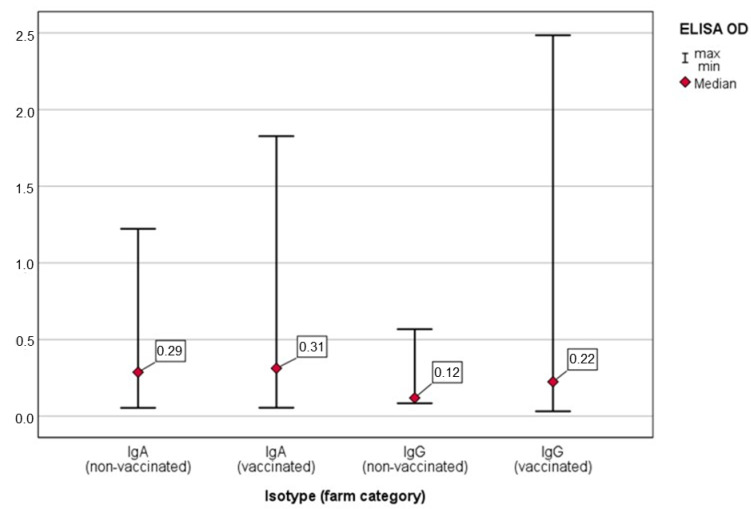
IgG and IgA ELISA OD value in the oral fluid samples from vaccinated and non-vaccinated farms (all pig categories).

**Figure 3 animals-11-02408-f003:**
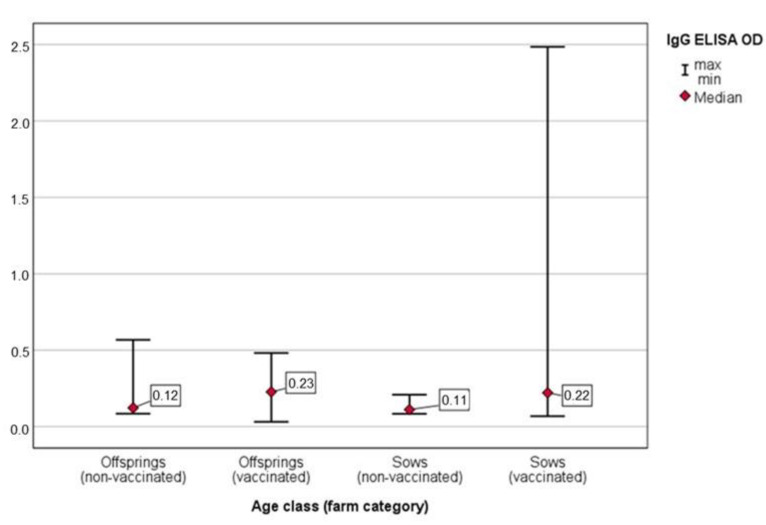
IgG ELISA OD value in the oral fluid samples from sows and offspring in vaccinated and non-vaccinated farms.

**Table 1 animals-11-02408-t001:** Number of oral fluid (OF) samples examined from vaccinated (V) and non-vaccinated (NV) pigs of three pig categories.

Pig Category	Farms 1 and 2 (V)	Farms 3 and 4 (NV)	Total
Weaners	13	16	29
Growers	11	19	30
Sows	13	9	22
Total OF samples	37	44	81

**Table 2 animals-11-02408-t002:** Median ELISA OD values (min–max) for anti-*Salmonella* IgA and IgG antibodies detected using IDEXX ELISA in oral fluid (OF) samples from four farms.

Isotype	Farm Category	Farm id.	Sows		Weaners		Growers		Offspring (Weaners and Growers)	All Animals	
IgG	V	1	0.13 (0.07–0.82)	(4) ^a^	0.30 (0.19–0.34)	(7)	0.23 (0.17–0.48)	(8)	0.26 (0.17–0.48)	0.24 (0.07–0.82)	(19)
		2	0.39 (0.07–2.49)	(9)	0.18 (0.03–0.32)	(6)	0.09 (0.08–0.09)	(2) ^b^	0.14 (0.03–0.32)	0.22 (0.03–2.49)	(17)
	NV	3	0.10 (0.09–0.14)	(6)	0.11 (0.09–0.28)	(7)	0.11 (0.09–0.57)	(10)	0.11 (0.09–0.57)	0.11 (0.09–0.57)	(23)
		4	0.17 (0.08–0.21)	(3)	0.14 (0.11–0.37)	(9)	0.13 (0.10–0.34)	(9)	0.14 (0.10–0.37)	0.14 (0.08–0.37)	(21)
	V	1 + 2	0.22 (0.07–2.49)	(13)	0.23 (0.03–0.34)	(13)	0.22 (0.08–0.48)	(10)	0.23 (0.03–0.48)	0.22 (0.03–2.49)	(36)
	NV	3 + 4	0.11 (0.08–0.21)	(9)	0.12 (0.09–0.37)	(16)	0.13 (0.09–0.57)	(19)	0.12 (0.09–0.57)	0.12 (0.08–0.57)	(44)
	V + NV	all farms	0.16 (0.07–2.49)	(22)	0.15 (0.03–0.37)	(29)	0.17 (0.08–0.57)	(29)	0.16 (0.03–0.57)	0.16 (0.03–2.49)	(80)
IgA	V	1	0.32 (0.31–0.66)	(4)	0.28 (0.09–0.51)	(7)	0.44 (0.21–1.83)	(8)	0.35 (0.09–1.83)	0.33 (0.09–1.83)	(19)
		2	0.53 (0.06–1.38)	(9)	0.08 (0.06–0.14)	(6)	0.12 (0.10–0.20)	(3)	0.10 (0.06–0.20)	0.15 (0.06–1.38)	(18)
	NV	3	0.28 (0.20–0.42)	(6)	0.07 (0.05–0.20)	(7)	0.16 (0.08–0.51)	(10)	0.13 (0.05–0.51)	0.18 (0.05–0.51)	(23)
		4	0.62 (0.17–1.08)	(3)	0.35 (0.20–0.47)	(9)	0.53 (0.25–1.22)	(9)	0.46 (0.20–1.22)	0.47 (0.17–1.22)	(21)
	V	1 + 2	0.46 (0.06–1.38)	(13)	0.14 (0.06–0.51)	(13)	0.43 (0.10–1.83)	(11)	0.22 (0.06–1.83)	0.31 (0.06–1.83)	(37)
	NV	3 + 4	0.30 (0.17–1.08)	(9)	0.23 (0.05–0.47)	(16)	0.47 (0.08–1.22)	(19)	0.27 (0.05–1.22)	0.29 (0.05–1.22)	(44)
	V + NV	all farms	0.35 (0.06–1.38)	(22)	0.20 (0.05–0.51)	(29)	0.43 (0.08–1.83)	(30)	0.25 (0.05–1.83)	0.30 (0.05–1.83)	(81)
IgG	Negative control ^c^		0.05 (0.05–0.06)	(4)	-	-	-	-			
IgA			0.54 (0.26–1.12)	(4)	-	-	-	-			

^a^ number of examined rope; ^b^ One OF sample was tested only for anti-*Salmonella* IgA; ^c^ four *Salmonella*-free sows housed in biosecure pens at the Animal and Plant Health Agency, UK.

**Table 3 animals-11-02408-t003:** Bacteriological results of samples collected from V and NV farms.

Farm Category	Pig Category	Farm	No. of Positive/Examined (%)		*p*	Serotype
V	Weaners	1	7/7	(100)	<0.01	4,5,12:i:- (7)
		2	0/6	(0.0)		-
	Growers	1	4/8	(50.0)	0.24	4,5,12:i:- (4)
		2	0/3	(0.0)		-
	Offspring ^a^	1	11/15	(73.3)	<0.001	4,5,12:i:- (11)
		2	0/9	(0.0)		-
	Sows	1	0/4	(0.0)	1	-
		2	2/9	(22.2)		Typhimurium (2)
	All pig categories	1	11/19	(57.9)	<0.01	4,5,12:i:- (11)
		2	2/18	(11.1)		Typhimurium (2)
NV	Weaners	3	7/7	(100)	1	Kedougou (7)
		4	9/9	(100)		4,5,12:i:- (9)
	Growers	3	10/10	(100)	0.47	Kedougou (10)
		4	8/9	(88.9)		4,5,12:i:- (8)
	Offspring	3	17/17	(100)	1	Kedougou (17)
		4	17/18	(94.4)		4,5,12:i:- (17)
	Sows	3	3/6	(50.0)	0.46	Kedougou (1); 4,5,12:i:- (2)
		4	3/3	(100)		4,5,12:i:- (3)
	All pig categories	3	20/23	(87.0)	0.61	Kedougou (18); 4,5,12:i:- (2)
		4	20/21	(95.2)		4,5,12:i:- (20)
V	Weaners	1 and 2	7/13	(53.8)	<0.01	4,5,12:i:- (7)
NV	Weaners	3 and 4	16/16	(100)		Kedougou (7); 4,5,12:i:- (9)
V	Growers	1 and 2	4/11	(36.4)	<0.01	4,5,12:i:- (4)
NV	Growers	3 and 4	18/19	(94.7)		Kedougou (10); 4,5,12:i:- (8)
V	Offspring	1 and 2	11/24	(45.8)	<0.001	4,5,12:i:- (11)
NV	Offspring	3 and 4	34/35	(97.1)		Kedougou (17); 4,5,12:i:- (17)
V	Sows	1 and 2	2/13	(15.4)	<0.05	Typhimurium (2)
NV	Sows	3 and 4	6/9	(66.7)		Kedougou (1); 4,5,12:i:- (5)
V	All pig categories	1 and 2	13/37	(35.1)	<0.001	Typhimurium (2); 4,5,12:i:- (11)
NV	All pig categories	3 and 4	40/44	(90.9)		Kedougou (18); 4,5,12:i:- (22)

^a^ Weaners and Growers.

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
