# Peer review of "Pilot Investigation of Anti-Salmonella Antibodies in Oral Fluids from Salmonella Typhimurium Vaccinated and Unvaccinated Swine Herds"

_animals, 2021, doi:10.3390/ani11082408_

Round 1

Reviewer 1 Report

The Salmonella control plan at the poultry level is implemented throughout Europe; however, this is not the case for the swine sector. The livestock sector requires control of this important zoonotic pathogen in live pigs. However, it is necessary to have cost-effective sample collection and testing methods that farmers can afford.  After including the indicated improvements, the article has the optimum level to be published. The results derived from this study provide very relevant information to search for cost-effective and efficient alternatives for the control of Salmonella in the field.

Author Response

The Salmonella control plan at the poultry level is implemented throughout Europe; however, this is not the case for the pig sector. The livestock sector demands the control of this important zoonotic pathogen in live pigs. However, cost-effective sample collection and analysis methods are necessary, capable of being assumed by farmers. That is why studies carried out at the field level, such as the one detailed in this research work, are very necessary for our sector.

It is a well-designed and well-written manuscript, with clear objectives and conclusions, so I only see a need for some data to be implemented.

Answer: The Authors thank the Reviewer for the positive comments on the paper and for the suggestions provided.

Firstly, taking into account that the work intends to use the ELISA technique as an alternative to the use of other official techniques for Salmonella monitoring, it is necessary to include in the manuscript data on the sensitivity of the test, the number of replications per sample performed, and coefficient of variation obtained.

Answer: The Authors thank the Reviewer for his comments. To our best knowledge, there are no validated commercially available immunoassays for the detection of anti-Salmonella antibodies in OF. Therefore, we decided to use and optimized the same IDEXX ELISA kit (validated only for serum and meat juice) used in numerous previous studies to focus on the detection of anti-Salmonella antibodies in porcine OF.

In our previous work (De Lucia et al, 2020, doi: 10.3389/fvets.2019.00489) we investigated the correlation of anti-Salmonella antibodies between serum and saliva samples collected from pigs.

Our previous results indicate that saliva samples had a sensitivity and a specificity of 86% and 80%, respectively when compared with ELISA results obtained from individual serum samples. Anti-Salmonella IgG antibodies in pig saliva were always detected at a lower level than in the matching serum samples and a significant correlation was found between individual Salmonella IgG in serum and saliva samples.

We also believed that including saliva negative controls (from Salmonella free animals) and the ELISA kit control, our results can be considered reliable, albeit not validated.

Information on the sensitivity and specificity of the IDEXX ELISA kit on MJ, sera, and saliva samples has been added at lines 159-164 and 166-168. We also added two new references (Farzan et al., 2007, doi:10.1017/S095026880600686; Vico et al., 2011, doi:10.1177/1040638711403432).

Information on the number of replicates and on the coefficient of variation has been added at lines 173-174 and at lines 195-197, respectively.

The manuscript indicates "On-farm surveillance of zoonotic diseases at the pre-harvest level is useful to help anticipate a public health problem long before it becomes a full-blown epidemic", but what benefits can it have compared to collecting feces/floor swabs that detect the presence of Salmonella on the farm? This point needs to be clarified. It would be necessary to include a paragraph in the introduction/discussion on how antigen detection can be useful for a Salmonella control/monitoring plan. What type of classification will be made in the herds? What differentiated management would be carried out in farms with a high IgG level compared to those that did not? etc.

In future related works, it would be very interesting to compare ISO (feces/floor swabs) versus IgG, because although it is true that they do not always correlate, it would be very interesting to be able to establish a monitoring protocol that allows making appropriate decisions as soon as possible at the field.

Answer: The Authors agree with the Reviewer’s comments. We have provided and answer to the questions raised below.

In our opinion, the OF may help in monitoring the disease dynamics in different age/production classes (as we investigated in our study), and potentially in different time points during the life of the animals (since stress free). Furthermore, the OF could be used to monitor vaccine efficacy and may be correlated/compared to the shedding rate.

Investigate the herd status is important to provide options for intervention strategies to reduce the prevalence from farm to fork. In particular, for Salmonella there is compelling evidence that the reduction of Salmonella intestinal carriage of live pigs and would help reduce the contamination pressure at the slaughterhouse. Consequently, Salmonella monitoring at farm-level even if more expensive then slaughterhouse surveillance programs should be included as part of comprehensive programs to reduce Salmonella contamination of pork. Monitoring the presence and prevalence of Salmonella in pig farms would allow for categorization of the risk before the animals are slaughtered. The detection of specific anti-Salmonella IgG in OF could allow evaluating the positive batch of pigs and to adopt specific prophylaxis measures at the slaughterhouse (e.g. end-of-day slaughter of high prevalence batches). However, further studies are needed to evaluate the correlation between Salmonella isolation and the presence of specific IgG in animals, in order to establish an effective monitoring protocol on farms.

These considerations and three new references (Alban et al., 2002, doi: 10.1016/s0167-5877(01)00270-7; Alban et al., 2010, doi:10.1111/j.1863-2378.2010.01367.x; Andres et al., 2015, doi: 10.1111/1541-4337.12137.) have been added in the Introduction (lines 60-70 and lines 74-76) and Conclusion paragraph (lines 354-359).

Minor comments:

Lines 22, 177: Review Salmonella throughout the manuscript; some italics are missing.

Answer: We apologize, the text was modified to address the Reviewer’s comment (line 22 and line 202).

Line 223: inactivated or attenuated?

Answer: We are sorry for the mistake. We corrected inactivated with attenuated (line 251).

Reviewer 2 Report

The manuscript titled "Pilot investigation of anti-Salmonella antibodies in oral fluids from Salmonella Typhimurium vaccinated and unvaccinated swine herds" carried out to investigate specific anti-Salmonella IgG and IgA antibodies levels in OF collected from pigs in farms that were vaccinated or not vaccinated against Salmonella Typhimurium, in comparison with the shedding of Salmonella in pooled faecal samples has been significantly improved by the authors. This study is important for the farmers on the field levels to control the Salmonella infection in the pig herds. Below are a few minor comments for the authors:

Line 176: Why was kit conjugate replaced with an anti-porcine IgA HRP conjugate? Isn't the IgA testing kit already available?

Line 181: How was the coefficient of variation (CV) was calculated? Please mention this in the manuscript.

Line 181 and 206: Please change "calculate" to "calculated".

Line 209: The IDEXX kit is not standardized for the saliva samples, but did the authors try to calculate the S/P ratio to compare the positives and negatives? If yes, were the results interpretable?

Line 389: The conclusion paragraph needs to be improved and the repetition of sentences should be avoided.

Author Response

The study entitled ‘Pilot investigation of anti-Salmonella antibodies in oral fluids from Salmonella Typhimurium vaccinated and unvaccinated swine herds’ carried out to investigate whether oral fluid can be a simple, cheap, and non-invasive alternative to serum or meat juice for diagnosis and surveillance of important pathogens in pigs is very well presented and its practical applicability is also the need of the hour.

Answer: The Authors thank the Reviewer for the positive comments on the paper.

I am curious regarding following queries:

Line 148: Why was kit conjugate replaced with an anti-porcine IgA HRP conjugate?

Answer: The IDEXX ELISA kit is designed for the detection of specific anti-Salmonella IgG and only an anti-porcine IgG conjugate is provided. We were also interested to detect the anti-Salmonella IgA as the IgA are the major immunoglobulin isotype in saliva produced by local plasma cells in the salivary gland. Therefore, the kit conjugate was replaced with an anti-porcine IgA HRP conjugate.

Table 2: Did the authors try to calculate the SP values? And what was the cut-off SP value for each kit?

Answer: The authors agree with the Reviewer’s comment that the calculation of SP and cut-off SP value would have been useful to establish. We have tried to calculate it but unfortunately with only the kit positive controls were not able to that. Ideally, we should have a saliva sample to use as a positive control for an accurate calculation, but this study design did not include bleeding of the animals. We believe that by including saliva negative controls (collected from Salmonella-free animals) and the ELISA kit control, our results can be considered reliable, albeit not validated. Ours study was a scoping exercise to see whether a validated ELISA test for serum was able to reliably detect anti-Salmonella antibodies in oral fluids from different groups of pigs.

However, in a previous study (De Lucia et al, 2020, doi: 10.3389/fvets.2019.00489), using the same IDEXX ELISA kit, we investigated the correlation of anti-Salmonella antibodies between serum and saliva samples collected from pigs. This second study cut-off and S/P values in sera and saliva samples at pen level and herd level were calculated and compared. Our result showed that the ELISA S/P ratio values for saliva samples were significantly lower than S/P values of the corresponding sera.

Table 2: The negative control OD value of IgA is 0.54±0.37 which indicates almost all the samples were negative. As mentioned in the manuscript, IgA ELISA is not reliable then what is the point of comparing OD values of pigs in different groups and ages?

Answer: The authors agree with the Reviewer’s comment, the detection anti-Salmonella IgA antibodies was found weak and non-specific, however for completeness we all the data were presented.

Table 2: Were all the vaccinated sow OF's positive for IgG?

Answer: All the vaccinated sows show high OD levels of IgG (mean 0.56; min-max: 0.16 to 1.06), and all of them (except for two) were negative for isolation of Salmonella.

Line 259 and 263: Please correct to Atkinson et. al.

Answer: We apologize to the Reviewer. The reference has been corrected (L. 287 and L. 291).

Table 5: Did the authors try to correlate the IgG levels and bacteological isolated Salmonella prevelance from the OF and feacal samples from same group of pigs? In other words, were there some samples where the IgG was negative but Salmonella was isolated and vice versa?

Answer: We were able to detect anti-Salmonella IgG in all the OF samples. Unfortunately, without a cut-off value, we can not tell which animal was positive or negative. As mentioned above our study was just a scoping exercise to see whether a validated ELISA test for serum could be adapted and able to detect anti-Salmonella antibodies in oral fluids from different groups of pigs.

This manuscript is a resubmission of an earlier submission. The following is a list of the peer review reports and author responses from that submission.

Round 1

Reviewer 1 Report

The Salmonella control plan at the poultry level is implemented throughout Europe; however, this is not the case for the pig sector. The livestock sector demands the control of this important zoonotic pathogen in live pigs. However, cost-effective sample collection and analysis methods are necessary, capable of being assumed by farmers. That is why studies carried out at the field level, such as the one detailed in this research work, are very necessary for our sector.

It is a well-designed and well-written manuscript, with clear objectives and conclusions, so I only see a need for some data to be implemented.

Firstly, taking into account that the work intends to use the ELISA technique as an alternative to the use of other official techniques for Salmonella monitoring, it is necessary to include in the manuscript data on the sensitivity of the test, the number of replications per sample performed, and coefficient of variation obtained.

The manuscript indicates "On-farm surveillance of zoonotic diseases at the pre-harvest level is useful to help anticipate a public health problem long before it becomes a full-blown epidemic", but what benefits can it have compared to collecting feces/floor swabs that detect the presence of Salmonella on the farm? This point needs to be clarified. It would be necessary to include a paragraph in the introduction/discussion on how antigen detection can be useful for a Salmonella control/monitoring plan. What type of classification will be made in the herds? What differentiated management would be carried out in farms with a high IgG level compared to those that did not? etc.

In future related works, it would be very interesting to compare ISO (feces/floor swabs) versus IgG, because although it is true that they do not always correlate, it would be very interesting to be able to establish a monitoring protocol that allows making appropriate decisions as soon as possible at the field.

Minor comments:

Lines 22, 177: Review Salmonella throughout the manuscript; some italics are missing.

Line 223: inactivated or attenuated?

Author Response

(The authors gave the same response as above.)

Reviewer 2 Report

The study entitled ‘Pilot investigation of anti-Salmonella antibodies in oral fluids from Salmonella Typhimurium vaccinated and unvaccinated swine herds’ carried out to investigate whether oral fluid can be a simple, cheap, and non-invasive alternative to serum or meat juice
for diagnosis and surveillance of important pathogens in pigs is very well presented and its practical applicability is also the need of the hour. I am curious regarding following queries:

Line 48: Why was kit conjugate replaced with an anti-porcine IgA HRP conjugate?

Table 2: Did the authors try to calculate the SP values? And what was the cut-off SP value for each kit?

Table 2: The negative control OD value of IgA is 0.54±0.37 which indicates almost all the samples were negative. As mentioned in the manuscript, IgA ELISA is not reliable then what is the point of comparing OD values of pigs in different groups and ages?

Table 2: Were all the vaccinated sow OF's positive for IgG?

Line 259 and 263: Please correct to Atkinson et. al.

Table 5: Did the authors try to correlate the IgG levels and bacteological isolated Salmonella prevelance from the OF and feacal samples from same group of pigs? In other words, were there some samples where the IgG was negative but Salmonella was isolated and vice versa? 

Author Response

The study entitled ‘Pilot investigation of anti-Salmonella antibodies in oral fluids from Salmonella Typhimurium vaccinated and unvaccinated swine herds’ carried out to investigate whether oral fluid can be a simple, cheap, and non-invasive alternative to serum or meat juice for diagnosis and surveillance of important pathogens in pigs is very well presented and its practical applicability is also the need of the hour.

Answer: The Authors thank the Reviewer for the positive comments on the paper.

I am curious regarding following queries:

Line 148: Why was kit conjugate replaced with an anti-porcine IgA HRP conjugate?

Answer: The IDEXX ELISA kit is designed for the detection of specific anti-Salmonella IgG and only an anti-porcine IgG conjugate is provided. We were also interested to detect the anti-Salmonella IgA as the IgA are the major immunoglobulin isotype in saliva produced by local plasma cells in the salivary gland. Therefore, the kit conjugate was replaced with an anti-porcine IgA HRP conjugate.

Table 2: Did the authors try to calculate the SP values? And what was the cut-off SP value for each kit?

Answer: The authors agree with the Reviewer’s comment that the calculation of SP and cut-off SP value would have been useful to establish. We have tried to calculate it but unfortunately with only the kit positive controls were not able to that. Ideally, we should have a saliva sample to use as a positive control for an accurate calculation, but this study design did not include bleeding of the animals. We believe that by including saliva negative controls (collected from Salmonella-free animals) and the ELISA kit control, our results can be considered reliable, albeit not validated. Ours study was a scoping exercise to see whether a validated ELISA test for serum was able to reliably detect anti-Salmonella antibodies in oral fluids from different groups of pigs.

However, in a previous study (De Lucia et al, 2020, doi: 10.3389/fvets.2019.00489), using the same IDEXX ELISA kit, we investigated the correlation of anti-Salmonella antibodies between serum and saliva samples collected from pigs. This second study cut-off and S/P values in sera and saliva samples at pen level and herd level were calculated and compared. Our result showed that the ELISA S/P ratio values for saliva samples were significantly lower than S/P values of the corresponding sera.

Table 2: The negative control OD value of IgA is 0.54±0.37 which indicates almost all the samples were negative. As mentioned in the manuscript, IgA ELISA is not reliable then what is the point of comparing OD values of pigs in different groups and ages?

Answer: The authors agree with the Reviewer’s comment, the detection anti-Salmonella IgA antibodies was found weak and non-specific, however for completeness we all the data were presented.

Table 2: Were all the vaccinated sow OF's positive for IgG?

Answer: All the vaccinated sows show high OD levels of IgG (mean 0.56; min-max: 0.16 to 1.06), and all of them (except for two) were negative for isolation of Salmonella.

Line 259 and 263: Please correct to Atkinson et. al.

Answer: We apologize to the Reviewer. The reference has been corrected (L. 287 and L. 291).

Table 5: Did the authors try to correlate the IgG levels and bacteological isolated Salmonella prevelance from the OF and feacal samples from same group of pigs? In other words, were there some samples where the IgG was negative but Salmonella was isolated and vice versa? 

Answer: We were able to detect anti-Salmonella IgG in all the OF samples. Unfortunately, without a cut-off value, we can not tell which animal was positive or negative. As mentioned above our study was just a scoping exercise to see whether a validated ELISA test for serum could be adapted and able to detect anti-Salmonella antibodies in oral fluids from different groups of pigs.